# Nobiletin Stimulates Adrenal Hormones and Modulates the Circadian Clock in Mice

**DOI:** 10.3390/nu16101491

**Published:** 2024-05-15

**Authors:** Conn Ryan, Yu Tahara, Atsushi Haraguchi, Yuanyuan Lu, Shigenobu Shibata

**Affiliations:** 1Laboratory of Physiology and Pharmacology, School of Advanced Science and Engineering, Waseda University, Tokyo 162-0056, Japan; connryan@akane.waseda.jp (C.R.); hmar.h@fuji.waseda.jp (A.H.); shibatas@waseda.jp (S.S.); 2Graduate School of Biomedical and Health Sciences, Hiroshima University, Hiroshima 734-0037, Japan; m236036@hiroshima-u.ac.jp

**Keywords:** nobiletin, circadian rhythm, peripheral tissue, functional food, phosphodiesterase, chrono-nutrition

## Abstract

Polymethoxyflavonoids, such as nobiletin (abundant in Citrus depressa), have been reported to have antioxidant, anti-inflammatory, anticancer, and anti-dementia effects, and are also a circadian clock modulator through retinoic acid receptor-related orphan receptor (ROR) α/γ. However, the optimal timing of nobiletin intake has not yet been determined. Here, we explored the time-dependent treatment effects of nobiletin and a possible novel mechanistic idea for nobiletin-induced circadian clock regulation in mice. In vivo imaging showed that the PER2::LUC rhythm in the peripheral organs was altered in accordance with the timing of nobiletin administration (100 mg/kg). Administration at ZT4 (middle of the light period) caused an advance in the peripheral clock, whereas administration at ZT16 (middle of the dark period) caused an increase in amplitude. In addition, the intraperitoneal injection of nobiletin significantly and potently stimulated corticosterone and adrenaline secretion and caused an increase in *Per1* expression in the peripheral tissues. Nobiletin inhibited phosphodiesterase (PDE) 4A1A, 4B1, and 10A2. Nobiletin or rolipram (PDE4 inhibitor) injection, but not SR1078 (RORα/γ agonist), caused acute *Per1* expression in the peripheral tissues. Thus, the present study demonstrated a novel function of nobiletin and the regulation of the peripheral circadian clock.

## 1. Introduction

Shift work and irregular lifestyles lead to a chronic misalignment of the circadian clock (jet lag), resulting in sleep disorders and lifestyle-related diseases such as obesity and hypertension [1]. The circadian clock is a cell-driven internal oscillating system with a 24 h cycle of clock genes, such as *Per1/2*, *Clock*, and *Bmal1*, which produce diurnal variations in various physiological functions. The central clock, located in the hypothalamus, controls the peripheral clock in the peripheral tissues. The circadian clock is strongly regulated by environmental factors, such as light and food. Chrono-nutrition research considers the timing of food intake and meal content [2]. Various functional food components have been reported to contribute to the circadian clock regulation. For example, caffeine, flavonoids, amino acids (such as L-serine), ω3 fatty acids, and triterpenoids have been reported as circadian clock-modulating agents [3]. Caffeine treatment before the sleep phase was found to delay mice and human circadian clocks; however, treatment in the early morning showed phase advancement [4,5]. Thus, the timing of the intake of clock-modulating supplements should be examined for translational research.

Flavonoids, which are abundant in fruit peels, have a variety of physiological activities. In particular, polymethoxyflavonoids such as nobiletin (abundant in *Citrus depressa*) and tangeretin have been reported to exhibit antioxidant, anti-inflammatory, anticancer, and anti-dementia effects [6]. In vitro screening assays have also shown that nobiletin affects the circadian clock with a prolonged period, increased amplitude, and phase modulation [7,8]. Nobiletin exhibits circadian clock regulation and anti-obesity effects, and shows agonistic action at retinoic acid receptor-related orphan receptors (RORs) α and γ, with a higher affinity for RORγ [8]. RORs are nuclear receptors that bind to the ROR element upstream of *Bmal1* as a transcription factor and promote *Bmal1* transcription. Nobiletin ameliorates the decreased clock amplitude and insulin secretion in beta cells derived from patients with type 2 diabetes [9,10]. Nobiletin inhibits triple-negative breast cancer growth via ROR agonist action and suppression of IκB/NF-κB signaling [11]. Nobiletin also modulates key signaling molecules (STAT3, NF-kb, COX-2, NRF2, and HO-1) involved in anti-inflammatory and anti-oxidative stress. While numerous studies have highlighted the beneficial effects of nobiletin as a functional nutrient for human health, there remains a gap in research concerning its acute impact on the circadian clock and hormonal fluctuations in vivo. In this study, we aimed to figure out the treatment-timing-dependent effect of nobiletin using in vivo PER2::LUC peripheral clock imaging in mice.

## 2. Materials and Methods

### 2.1. Ethical Approval

The experimental procedures were conducted with the approval of the Animal Care Committee at Taisho Pharmaceutical Co., Ltd. (Tokyo, Japan), in accordance with the company’s guidelines for the Care and Use of Laboratory Animals (AN13437-Z00), or with the approval of the Committee for Animal Experimentation of the School of Science and Engineering at Waseda University (permission protocol nos. 2013-A059, 2013-A060). The experiments were performed in accordance with the law (No. 105) and notification (No. 6) of the Japanese government.

### 2.2. Animal Housing and Nobiletin Treatment

Male C57BL/6J mice (7 weeks old) were purchased from Charles River Japan (Kanagawa, Japan) and used for the serum hormone assay and RT-PCR. Male ICR mice were purchased from Tokyo Laboratory Animals Science Co., Ltd. (Tokyo, Japan), and female PERIOD2::LUCIFERASE (PER2::LUC) knock-in mice (ICR background) bred at Waseda University were used for blood glucose measurements and in vivo imaging assays, respectively. The mice were all housed under conditions of controlled temperature (23 °C ± 3 °C), humidity (50% ± 20%), and lighting (lights on from 07:00 to 19:00 at Taisho Pharmaceutical Co., Ltd. or from 08:00 to 20:00 at Waseda University). In the 24 h cycle, Zeitgeber time 0 (ZT0) was designated as the time when the lights were turned on, and ZT12 was designated as the time when the lights were turned off. All the mice were given access to food and tap water ad libitum, and nobiletin (10–100 mg/kg; purchased from FUJIFILM Wako Pure Chemical Corporation, Osaka, Japan) was administered by intraperitoneal (i.p.) injection in the vehicle (10 mL/kg body weight) containing 0.5% carboxymethyl cellulose (CMC). Rolipram (PDE4 inhibitor, FUJIFILM Wako Pure Chemical Corporation) and SR1078 (RORα/γ agonist, FUJIFILM Wako Pure Chemical Corporation) were also used for the experiment at 10 mg/kg in 0.5% CMC [12]. Mice were euthanized by rapid decapitation for whole blood samples.

### 2.3. In Vivo Bioluminescence Imaging

An in vivo imaging system (IVIS) (Caliper Life Sciences, Hopkinton, MA, USA) was used to monitor the PER2::LUC activity rhythm waveform in the peripheral tissues (kidney, liver, and submandibular gland) of the mice. A previously developed protocol and analytical method to monitor bioluminescence oscillations in peripheral tissues was used [13]. Briefly, the mice were anesthetized with isoflurane (FUJIFILM Wako Pure Chemical Corporation, Osaka, Japan) and enriched with oxygen. Mice were anesthetized and injected subcutaneously on the back and near the neck with D-luciferin potassium salt (FUJIFILM Wako Pure Chemical Corporation, Osaka, Japan) at a dose of 15 mg/kg body weight. Images were taken using an IVIS, in the dorsal-up position for the kidney 8 min after injection and in the ventral-up position for the liver and the submandibular gland 10 min after injection. Images were obtained six times/day at 4 h intervals (ZT3, 7, 11, 15, 19, and 23) (Figure 1 and Figure 2). The mice were returned to their home cages after each imaging procedure and quickly recovered from isoflurane anesthesia. Bioluminescence emitted from the kidneys and liver was calculated automatically using Living Image 3.2 software (Caliper Life Sciences). The average photon/min value for the six time points from each day was designated as 100%, and the bioluminescence rhythm for the entire day was expressed as a percentage of each set of six time points for individual organs. The peak phase and amplitude of the normalized data were determined using a single cosinor procedure program (Acro.exe version 3.5 [14]).

### 2.4. Blood Assays

Serum corticosterone, adrenaline, and adreno-corticotropic hormone (ACTH) levels, and corticotropin releasing factor (CRF) in the hypothalamus, were measured using enzyme-linked immunosorbent assay (ELISA). A corticosterone ELISA kit (Cayman Chemical Company, Ann Arbor, MI, USA), an adrenaline ELISA kit (Abcam, Cambridge, UK), an ACTH ELISA kit (Phoenix Pharmaceuticals Inc., Burlingame, CA, USA), and a CRF ELISA kit (Yanaihara Institute Inc., Shizuoka, Japan) were used. Assays were conducted according to the manufacturer’s instructions. Blood samples were collected from the mice at ZT4, 5, and 7 after the administration of nobiletin at ZT3 (10 mg/kg and 100 mg/kg, in Figure 3A,B). Blood glucose levels were measured using a Glucose PILOT kit (Aventir Biotech, LLC, Carlsbad, CA, USA). The range of blood glucose that could be precisely measured with the kit was 20–600 mg/dL. Before measurement, we administered nobiletin (100 mg/kg body weight) to the mice, after 4 h fasting. Blood was collected from the tail vein of each mouse for measurement. The measurements were conducted at 0, 15, 30, 45, 60, and 90 min after reagent administration in Figure 3C. Samples were collected one hour after nobiletin treatment (50 or 100 mg/kg; Figure 3D,E), or 2 or 4 h after treatment in Figure 3F.

### 2.5. Real-Time RT-PCR

TRIzol (Life Technologies, Carlsbad, CA, USA) was used to extract RNA from peripheral tissues. Real-time reverse transcription PCR (RT-PCR) was performed using the One-Step SYBR RT-PCR Kit (Takara Bio Inc., Shiga, Japan). Primers were used as described previously [15] and PCR was performed using a Piko Real PCR system (Thermo Fisher Scientific, Waltham, MA, USA). From the comparison of two housekeeping gene expressions (*Gapdh* and *Tbp*) in the current samples among groups, the expression levels of target genes were normalized to those of *Gapdh*. We used the 2^−ΔΔCt^ method for data analysis and performed a melt curve analysis on each primer to identify any non-specific products.

### 2.6. Kinase Assay

Enzyme assays for various subtypes of PDE (Table 1 and Figure 4) were conducted by Eurofins Panlabs Discovery Services Taiwan, Ltd. (New Taipei City, Taiwan). In the assays, five or six concentrations of nobiletin over five or six orders of magnitude were examined in experiments conducted in duplicate to determine the IC50 (concentration of nobiletin exerting half-maximal inhibition of control-specific activity). IC50 values were determined by nonlinear least squares regression analysis using MathIQTM (ID Business Solutions Ltd., Woking, UK). FAM-cAMP, FAM-cGMP (0.1 μM), and 100 μM [^3^H]cGMP + cGMP were used for the substrate in the human recombinant insect Sf9 cells or Bovine retinal rod outer segments depending on the PDE subtype. The spectrofluorometric quantitation of fluorescein-AMP-IMAP, fluorescein-GMP-IMAP, or [^3^H] guanosine was used depending on the PDE subtype. A known agonist of each PDE was used as the positive control.

**Table 1 nutrients-16-01491-t001:** IC50 doses of PDE inhibitory activity of nobiletin by kinase assay.

Target PDE	IC_50_ Dose [μM]
PDE1A	-
PDE2A	9.91
PDE3A	16.0
PDE4A1A	5.23
PDE4B1	6.03
PDE5A	14.8
PDE6	74.5
PDE7A	11.9
PDE7B	20.4
PDE8A1	-
PDE9A2	-
PDE10A2	1.51
PDE11A4	31.0

### 2.7. Statistical Analyses

Data were expressed as mean ± standard error of the mean (SEM) and were statistically analyzed using GraphPad Prism version 6.03 (GraphPad Software, San Diego, CA, USA). We determined whether the data showed normal or non-normal distribution and equal or biased variation using the D’Agostino–Pearson test/Kolmogorov–Smirnov test and the F value test/Bartlett’s test, respectively. Parametric analysis was conducted using one-way or two-way ANOVA with a Tukey test, Sidak test, or Student’s *t*-test for post hoc analysis. Non-parametric analysis was performed using the Kruskal–Wallis test with Dunn’s test or the Mann–Whitney test for post hoc analysis. Statistical significance was set at *p* < 0.05.

## 3. Results

### 3.1. Nobiletin-Induced Circadian Clock Changes Showed Time-of-Day Dependence

We investigated the effects of nobiletin administration on the phase and amplitude of peripheral PER2::LUC rhythms using in vivo whole-body imaging (Figure 1 and Figure 2). Phase response curves (PRC) and amplitude changes were analyzed after continuous 3-day treatment with the control or nobiletin (100 mg/kg, i.p.) at different treatment timings (Figure 1 and Figure 2). Nobiletin administration at ZT4 showed the phase advancement of PER2::LUC rhythms in the kidney and submandibular gland compared with the control treatment (Figure 2B,C). The liver clock was phase-advanced by nobiletin treatment at ZT12 (Figure 2B,C). The amplitude was increased by nobiletin treatment at ZT16 in all three tissues, but decreased by treatment at ZT0 or 4 in the kidney and ZT12 in the liver (Figure 2B,D). These results suggest that daily nobiletin treatment strongly changes the peripheral clock phase and amplitude in a time-of-day-dependent manner.

### 3.2. Nobiletin Stimulated Adrenal Hormones and Peripheral Per1 Expression

PER2::LUC changes in the kidney by nobiletin treatment at ZT0 and 4 are similar to the changes caused by physiological stress treatment [15]. Physiological stress-induced circadian clock changes are mediated by adrenal hormones [15]. Thus, we next measured the serum adrenal hormone levels after nobiletin treatment (Figure 3). Nobiletin administration with a higher concentration (100 mg/kg), but not with a lower concentration (10 or 50 mg/kg), caused an increase in the serum corticosterone levels 1 h after i.p. administration (Figure 3A,D). The levels of corticosterone and adrenaline showed a similar time course by measuring each value at 1, 2, and 4 h after 100 mg/kg nobiletin i.p. injection (Figure 3B). An increase in corticosterone and adrenaline secretion is reported to cause gluconeogenesis [16]. By measuring blood glucose levels after nobiletin administration (100 mg/kg), we found that nobiletin also increased blood glucose levels (Figure 3C). In the hypothalamic–pituitary–adrenal axis, CRF in the hypothalamus and serum ACTH did not change significantly an hour after injection (Figure 3D), suggesting the direct effect of nobiletin on the adrenal grand. Corticosterone and adrenaline secretion initiate *Period* gene expression in peripheral tissues [17]. *Per1* mRNA in the kidney, liver, and stomach was upregulated by nobiletin administration, and this effect was diminished 4 h after treatment (Figure 3E,F). *Per2* expression decreased significantly in the liver, suggesting a tissue-specific effect of nobiletin.

### 3.3. Nobiletin Possessed Inhibitory Effects on the PDE Family, Especially PDE4 and 10

Previous studies have shown that nobiletin inhibits PDE3 [18]. However, the inhibitory effects of nobiletin on the rest of the PDEs have yet to be revealed. In addition, the administration of the PDE4 inhibitor rolipram caused an increase in the circulating levels of both corticosterone and adrenaline in male mice [19]. Therefore, we measured the inhibitory effects of nobiletin on the PDE family (PDE1–PDE11) using kinase assays. Various PDE-inhibitory effects of nobiletin were observed (Table 1). Although the nobiletin effects were lower compared with the positive control, the IC50 values of nobiletin for PDE4A1A, 4B1, and 10A2 were more evident than those for the other PDEs (Table 1 and Figure 4A). We next examined whether nobiletin administration induced effects similar to those of a PDE4 inhibitor in mice. Corticosterone and adrenaline secretion and *Per1/2* mRNA changes in the stomach and lung were detected after treatment with rolipram but not SR1078 (Figure 4B,C). Although nobiletin activates RORs and enhances *Bmal1*, *Npas2*, and *Dec1* expression in aged mice fed a high-fat diet [20], *Bmal*1 expression did not change significantly after a single treatment with nobiletin or SR1078 in our study (Figure 3E,F and Figure 4C).

## 4. Discussion

To the best of our knowledge, this is the first study to show that nobiletin produced timing-dependent phase shifts and amplitude changes in peripheral clocks. We also found that nobiletin stimulates adrenaline and corticosterone secretion, which in turn may contribute to the regulation of the circadian clock.

The PRC of the peripheral clock induced by nobiletin was similar to the results of previous studies on the PRC of dexamethasone treatment [21,22] or restraint stress-induced peripheral clock entrainment [15]. In the kidney, nobiletin treatment at ZT4 induced a phase advance, treatment at ZT20 induced a phase delay, and treatment at ZT0 induced an amplitude decrease (Figure 2). The amplitude reduction by nobiletin at ZT0 was a singular phenomenon due to the intercellular phase shift with both advance and delay, which was also observed with restraint stress treatment [15]. The amplitude increase induced by nobiletin treatment at ZT16 or ZT20 might be through the ROR pathway, since RORs are highly expressed at these times [8]. Since we thought that the PRC of nobiletin was similar to that of stress treatment, we next found a novel effect of nobiletin on the adrenal stress hormones (Figure 3). However, nobiletin might regulate peripheral clocks through multiple pathways. Chronic nobiletin treatment enhances the amplitude or expression of clock genes in the peripheral tissues and cortex of young and older high-fat-diet-treated [8] or Alzheimer’s disease model (APP/PS1) mice [23,24]. However, *Bmal1* and other clock gene mRNA rhythms were not consistently altered by chronic nobiletin treatment in the cortex of normal WT mice [23,24]. No studies have shown acute changes in clock gene expression following nobiletin treatment. While employing only one housekeeping gene in the present study may be considered a potential limitation, we observed acute *Per1* mRNA induction in peripheral tissues after 1–2 h of rolipram or nobiletin administration, but not by an ROR agonist (Figure 3 and Figure 4). *Per2* was downregulated by nobiletin in the liver, and *Bmal1* did not change with nobiletin, rolipram, or SR1078 (Figure 3 and Figure 4). These results suggest that nobiletin acutely regulates the peripheral clock through a pathway different from that of ROR. Rolipram causes an increased expression of *Per1* in fibroblasts via elevated cAMP and the activation of cAMP response element-binding protein (CREB) [18,25], suggesting that the PDE4 inhibitory effect of nobiletin may directly regulate peripheral clock gene expressions. In addition, elevated blood corticosterone and adrenaline levels due to nobiletin administration may indirectly cause gene expression changes of the peripheral clock. The secretion of corticosterone and adrenaline by stress and exercise or the intraperitoneal administration of these hormones directly modulates the rhythm of the clock gene expression in the liver and other organs [15,26]. Corticosterone regulates *Per1* and *Per2* expression via glucocorticoid receptors and adrenaline via the cAMP-CREB pathway [17]. Pharmacodynamics of nobiletin in a previous study indicated the highest plasma concentration of nobiletin during 2–4 h after nobilin treatment in rats [27], supporting the direct effect of nobiletin on the hormonal secretions and clock gene expressions in the current study. Thus, the PRC of nobiletin in this study was the result of multiple functions of nobiletin on the peripheral clocks.

The PDE kinase assay revealed that nobiletin exhibited a more specific inhibitory effect on PDE4A1A, 4B1, and 10A2 with ≦6.03 μmol/L of IC50 (Table 1 and Figure 4). A previous study showed that nobiletin inhibits PDE3 (IC50 = 10.4 μmol/L) [18]. Sudachitin (from Citrus sudachi), a polymethoxyflavonoid, has also been reported to inhibit PDE1C and PDE4B [28]. In this study, nobiletin also inhibited PDE4B (IC50 = 4.7 μmol/L). The inhibition of PDE leads to an increase in intracellular cAMP, the activation of cAMP-dependent protein kinase, and the activation of anti-inflammatory pathways, such as NF-kb [29]. Previous studies have reported that the intraperitoneal administration of rolipram (40–200 μg/kg), an inhibitor of PDE4, increases corticosterone and adrenaline secretion [19]. Furthermore, denbufylline, a PDE4 inhibitor, promotes corticosterone secretion by altering the activity of the pituitary via the HPA system [30]. However, another study found that even rolipram and denbufylline treatment of the isolated hypothalamic in vitro caused significant production of CRF, and that the intraperitoneal injection of denbufylline but not rolipram to mice produced CRF induction in vivo [31]. These results suggest that the PDE4 inhibitor could have a direct effect to the adrenal gland or an indirect affect through HPA axis activation. Indeed, the current study did not confirm the increase in CRF or ACTH by nobiletin administration (Figure 3D). In addition, a low dose of rolipram treatment (1.6 and 8 μg/kg) did not produce serum corticosterone increase [31], similar to the current result, where 100 mg/kg but not 10 mg/kg or 50 mg/kg of nobiletin caused corticosterone production (Figure 3). Our current data at least suggest that a high dose of nobiletin stimulates adrenal hormone secretion, which might be through PDE4 inhibition.

This study has the following limitations: Although we evaluated the acute circadian clock response to nobiletin, we did not assess the effects of long-term administration (lasting more than one week). Additionally, while nobiletin affects peripheral clocks through various pathways, we have not thoroughly evaluated which pathways mediate the observed responses in in vivo imaging. Furthermore, it remains to be investigated whether the circadian clock response to nobiletin occurs in humans. These aspects will require further investigation in future studies.

## 5. Conclusions

In summary, we explored a novel mechanism of nobiletin-induced circadian clock regulation and the time-dependent effects of nobiletin in mice. Our results suggest that nobiletin administration at the beginning of the active period might be the best time to enhance clock oscillation. Since the circadian clock has a unique aspect, in which the response of entrainment stimulation depends on treatment timing, our data are beneficial for translational research on nobiletin supplementation in humans. Nobiletin is utilized as a functional food ingredient with antioxidant, anti-inflammatory, anticancer, and anti-dementia effects. The findings of this study contribute valuable insights for product development considering the effects of nobiletin on the circadian clock.

## Figures and Tables

**Figure 1 nutrients-16-01491-f001:**
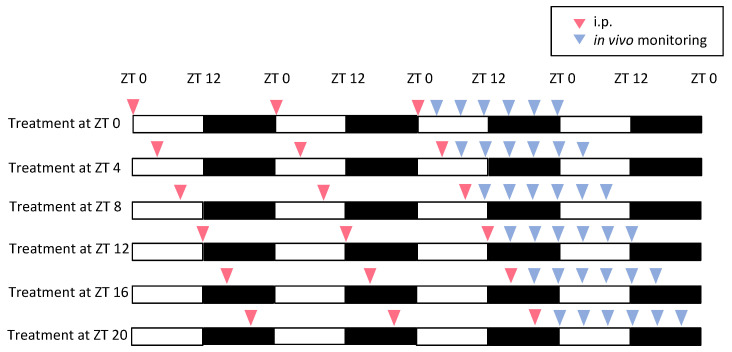
Experimental protocol for in vivo imaging. PER2::LUC female mice were treated with control (0.5% CMC) or nobiletin (100 mg/kg) at the same time points (ZT0, 4, 8, 12, 16, or 20) for three consecutive days, and then measured for peripheral clocks by in vivo whole-body imaging to examine phase–response curves (*n* = 4–5 in each).

**Figure 2 nutrients-16-01491-f002:**
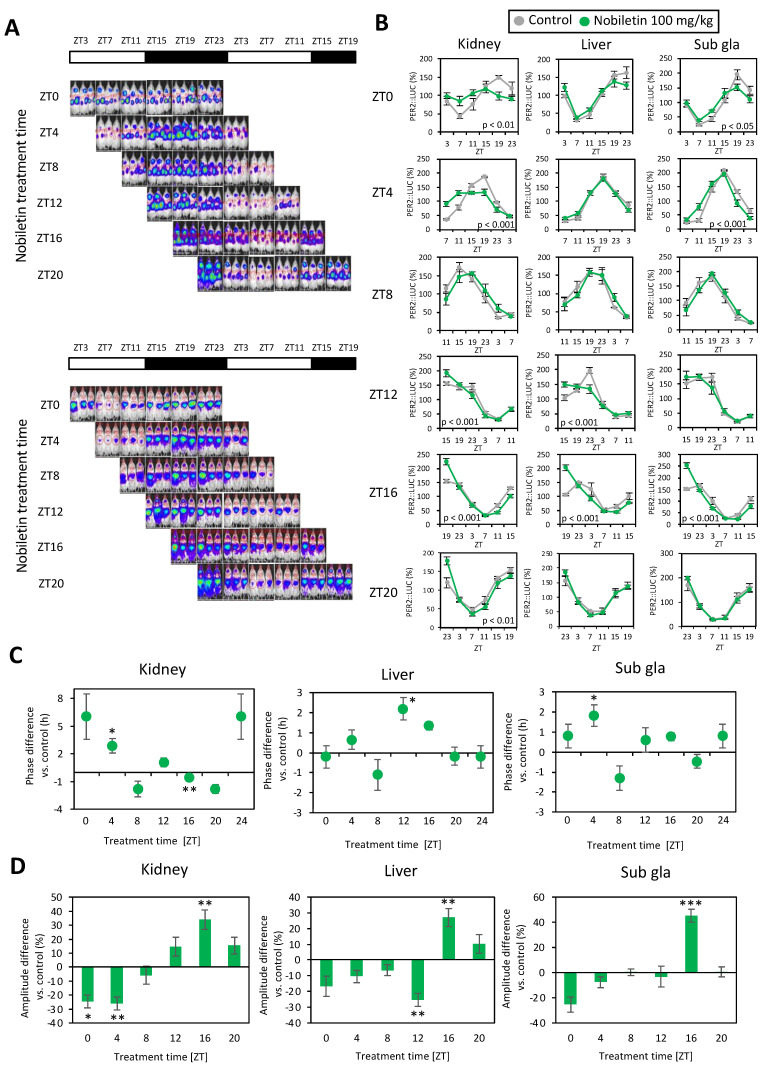
Time-of-day dependence of peripheral PER2::LUC changes in nobiletin injection. (**A**) Representative photo images of in vivo PER2::LUC bioluminescence in the kidney, liver, and submandibular gland (sab gla). (**B**) Waveforms of PER2::LUC rhythm in each tissue at each treatment time. (**C**,**D**) Phase–response curves or amplitude–response curves of nobiletin injections in peripheral PER2::LUC rhythm. Phase or amplitude differences between control and nobiletin treatment are indicated for each treatment timing in each tissue (*n* = 4–5 in each). * *p* < 0.05, ** *p* < 0.01, *** *p* < 0.001 vs. control at each time point by Student’s *t*-test.

**Figure 3 nutrients-16-01491-f003:**
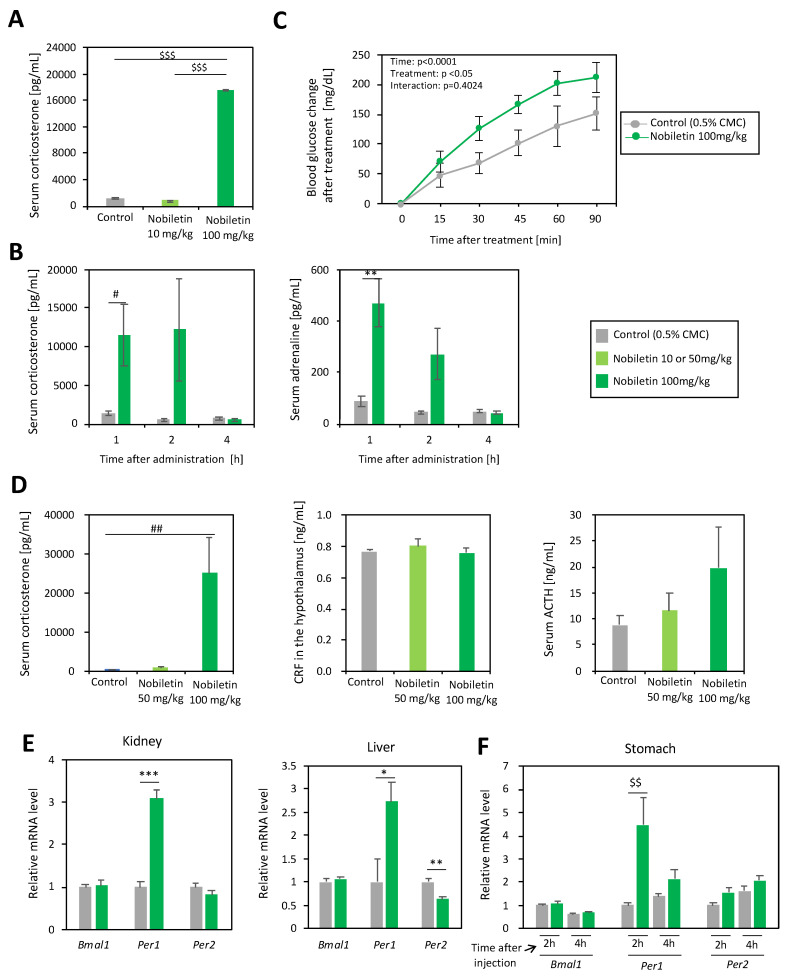
Nobiletin stimulated adrenaline and corticosterone secretion, and induced acute clock gene expression changes. (**A**–**C**) Mice received intraperitoneal injections of control (0.5% CMC) or nobiletin (10 or 100 mg/kg) at ZT3. (**A**) Serum corticosterone at 1 h after treatment (*n* = 6 in each). $$$ *p* < 0.001 by Tukey’s multiple comparisons test. (**B**) Serum corticosterone and adrenaline at 1, 2, and 4 h after treatment (*n* = 4 in each). # *p* < 0.05 vs. control at each time point by Mann–Whitney test; ** *p* < 0.01 vs. control at each time point by Student’s *t*-test. (**C**) Blood glucose changes after treatment (*n* = 9 in each). Statistics of 2-way repeated ANOVA are indicated on the graph. (**D**) Serum corticosterone, CRF in the hypothalamus, and serum ACTH at an hour after treatment (*n* = 5 in each). ## *p* < 0.01 vs. control by Mann–Whitney test. (**E**) Relative clock gene mRNA levels in the kidney (left panel) or liver (right panel) at an hour after treatment (*n* = 5 in each). * *p* < 0.05, ** *p* < 0.01, *** *p* < 0.001 vs. control by Student’s *t*-test. (**F**) Relative clock gene mRNA levels in the stomach at 2 or 4 h after treatment (*n* = 5 in each). $$ *p* < 0.01 by Tukey’s multiple comparisons test.

**Figure 4 nutrients-16-01491-f004:**
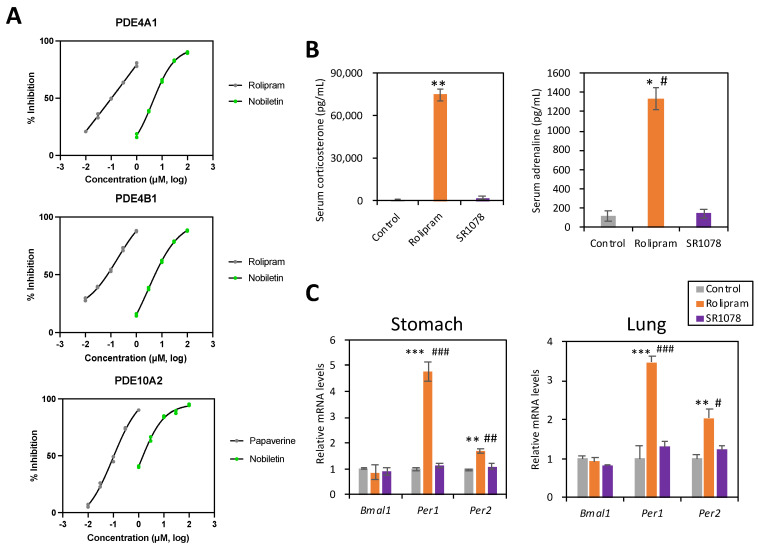
PDE kinase assays and the acute clock gene expression changes by PDE4 inhibitor or ROR α/γ agonist. (**A**) Inhibitory effect of nobiletin on PDE4A1, 4B1, and 10A2. Various concentrations of nobiletin (1–100 μM) were used for IC50 determination (Table 1). Values represent the mean of two experiments. Positive controls were rolipram for PDE4A1 and 4B1 and papaverine for PDE10A2. (**B**) Mice received control (0.5% CMC, i.p.), rolipram (PDE4 inhibitor, 10 mg/kg in 0.5% CMC, i.p.), or SR1078 (RORα/γ agonist, 10 mg/kg in 0.5% CMC, i.p.) at ZT3. Serum corticosterone and adrenaline levels 1 h after treatment (*n* = 4 per group). * *p* < 0.05, ** *p* < 0.01 vs. control; # *p* < 0.05 vs. SR1078 by Dunn’s multiple comparison test. (**C**) Relative clock gene mRNA levels in the stomach and lungs 2 h after treatment (*n* = 5 per group). ** *p* < 0.01, *** *p* < 0.001 vs. control; # *p* < 0.05, ## *p* < 0.01, ### *p* < 0.001 vs. SR1078 by Tukey’s multiple comparisons test.

## Data Availability

The data used in this study are the property of the company and will not be released to the public. However, the data will be provided to researchers upon request for research purposes.

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
