# Peer review of "Nobiletin Stimulates Adrenal Hormones and Modulates the Circadian Clock in Mice"

_nutrients, 2024, doi:10.3390/nu16101491_

Round 1

Reviewer 1 Report

Comments and Suggestions for Authors

The study explores the time-of-day administration effects of nobiletin, a polymethoxyflanoid found abundantly in Citrus depressa, on circadian clock. It clearly demonstrated that changes in the circadian clock of peripheral clocks depend on time-of-day administration. Also, it proposes a mechanist pathway for the findings that involves the hypothalamus adrenal axis and action on the phosphodiesterase. It was well-designed. It brings a novelty. The only weakness was the conclusion, it should state a broadly use of nobiletin.  And, it should explain the pharmacodynamics of the drugs, since the effect present could be due to the washout time in the chronic treatment.

Minor:

Line 51 – Bmal1 should be in italics.  

Line 82 – Specify the sources of Rolipram and SR1078

Line 103 – Was the Acro.exe program designed by Dr Refinetti? It should be referred if so.

Line 117 – Remove an extra period.

Reviewer 2 Report

Comments and Suggestions for Authors

The manuscript by Ryan et al. presents evidence regarding the effects of nobiletin on the circadian clock, describing its role in phase advancement and amplitude modulation. The study explores potential mechanisms involving adrenal hormones and clock gene expression, further supported by the inhibitory effects of nobiletin on PDE enzymes, particularly PDE4 and PDE10.

My evaluation is overall positive, especially considering the numerous experiments supporting the authors' thesis. However, I believe that some aspects, mainly in the introduction and discussion, need to be further explored:

-       The introduction, while concise, lacks the strength to engage the reader fully. Strengthening it with a more expansive description
of some molecular mechanisms of nobiletin that support the manuscript's aim would enhance its appeal and clarity.

-       The use of only one housekeeping gene in the real-time PCR data analysis may compromise statistical validity. It is advisable to incorporate at least two, preferably three, housekeeping genes to ensure accuracy and reliability.

-              The study offers valuable insights into the molecular mechanisms underlying nobiletin's impact on circadian rhythms in mice, but its clinical relevance to human health remains uncertain. Further research, potentially involving human subjects or translational models, is necessary to elucidate the applicability of these findings to human populations. A more comprehensive discussion of this aspect is warranted.

-       The discussion section could benefit from a more thorough examination of the study's limitations, including potential confounding factors, methodological constraints, and alternative interpretations of the results. For example, the study primarily focuses on acute effects following nobiletin administration. Long-term effects and potential adverse outcomes, especially with prolonged or repeated exposure, are not explored. Considering the interest in nobiletin as a dietary supplement, investigating its chronic effects is essential for a comprehensive understanding of its safety and efficacy.

Comments on the Quality of English Language

Minor editing of English language required.

Round 2

Reviewer 2 Report

Comments and Suggestions for Authors

The authors have addressed my comments, making appropriate modifications to various sections of the manuscript.